# Maybe a Few Considerations in Reinforcement Learning Research?

Kamyar Azizzadenesheli [1]

## Abstract

Recent advances in computation power accessible to machine learning researchers has sparked the flurry of research interest in large scale Reinforcement Learning (RL). However, this interest is facing RL unique characteristics; Any theoretical study as well as empirical investigation of any innovative RL algorithm requires an extensive amount of experts' time and imposes intolerably massive computation cost. These challenges are even more vital in RL due to the significantly limited dedicated resources, such as human-time and computation resources, to RL research. These hardnesses and enormous costs impel an immediate consideration in both guiding our research directions and allocating our resources to advance RL in practice. We devote this paper to discuss a set of essential issues that are necessary to be addressed in order to provide greater supervisions in conducting future RL studies.

## 1. Introduction

RL as the study of sequential decision making under uncertainty is the core of data-driven approaches in real-world applications, abounding from healthcare (Thompson, 1933) and recommendation systems (Li et al., 2010) to autonomous vehicles (Lange et al., 2012), dialogue systems (Singh et al., 2000), and robotics (Argall et al., 2009). This wide range of applications has made RL an important topic of interest for many decades. Researcher have made many principled and heuristic contribution in this field  (Lai & Robbins, 1985; Lattimore & Szepesvári, 2018; Brafman & Tennenholtz, 2002; Auer et al., 2002; Jaksch et al., 2010; Kocsis & Szepesvári, 2006; Sutton, 1990) and have shed lights on the essence of RL.

Recent advances in computation capability and techniques in deep learning have allowed the machine learning community to provide significant progress in RL using deep learning, known as deep RL (DRL). Among the first best-publicized methods, deep Q-network (DQN) (Mnih et al., 2015) tackles video games in Atari Learning Environment (Bellemare et al., 2013), AlphaGo addresses board games (Silver et al., 2016), and Deep Visuomotor Policies to approach robotic problems (Levine et al., 2016).

*These all are promising for further improvements in RL.*

Despite all the principled and empirical advances in RL, this problem is shockingly barely understood and deserves much more attention from theorists and practitioners.

*Maybe a sad news*:  The study of RL indicates that this problem is hard from information theoretic point of view (Jaksch et al., 2010; Li et al., 2015; Ortner & Ryabko, 2012; Jiang et al., 2017). They suggest that there are many settings that there might not even exist any RL method which achieves our desired performance. Another bad news, among all the good news: any innovation in either theory or practice requires a massive amount of effort.

In this short paper, we argue how the lack of theoretical understanding and rigor can have a drastic contribution to the lack of broad applicability of current RL methods in real-world applications. We argue that while we do not clearly understand how to approach an RL problem from the very first and elementary principles, we may not satisfyingly claim significant progress in RL. [1]

Let us rethink a bit about the following topics and issues:

## 2. Policy Gradient

Policy gradient (PG) is a prominent approach in RL where we directly learn our policies using gradient methods (Aleksandrov et al., 1968; Rubinstein, 1969). The preliminary PG methods, despite generality, mainly suffer from high variance estimation in improvement direction, resulting in poor empirical performance. Recent development shows how to reduce the variance of gradient estimation and guarantee local improvement (Kakade & Langford, 2002; Schulman et al., 2015).

[1]UC, Irvine, Caltech, *kazizzad@caltech.edu*.

[1]*Disclaimer*: Most of our statements are a bit exaggerating to bold the point we try to make. We also do not advocate these discussions are the most relevant ones (in fact, we are open to feedback) but rather a way to update the readers' belief, credible interval, and hopefully confidence set, all with high probability.

These methods later have been successfully evaluated on control domains (Lillicrap et al., 2015; Schulman et al., 2017).

*In general, it is a non-convex problem as it is formulated.*

Optimizing the performance of an RL agent in the policy space is mainly non-convex, and some of the current principled PG methods are guaranteed to provide local improvement. But they neither guaranteed to converge to a reasonable solutions nor guaranteed (yet) to behave efficiently when sample complexity is our concern.

Clearly, the final performance of such methods depends heavily on their initialization as it is obviously should be the case for non-convex problems when we deal with them for finite time steps. This phenomenon is also empirically observed in Henderson et al. (2018). At the current stage of the literature, we do not know how to efficiently do PG.

*What can maybe help us to move forward?*

These days, empirical studies are the leading figure of the PG literature. In this literature, we mainly evaluate the significance of new methods on a small set of toy environments. In this scientific atmosphere, there are also experimental and theoretical works which dedicate their efforts in the study of PG approaches from the first principles to shed lights on how to design efficient PG methods. Since these works mainly do not aim to climb higher on the ladder of leader-board scores, they are in high danger of being doomed by reviewers. They will be left unappreciated if they do not outperform the current state of the art methods which are designed to overfit to a set of toy environments, *refer to section 3*.

While empirical studies provide significantly useful pieces of knowledge and intuition in the design of efficient RL algorithm, theoretical understand will adjust our general direction. In particular, in PG, where we barely understand the behavior of our methods and mainly hope for our best shots, principled study can help to design better and robust algorithms. We think that successful methods in PG are possible if we pave the road for theoretical advancement and appreciate empirical studies which do not aim to climb the leader-board, instead, provide principled studies on the design of efficient PG methods.

*While competing against state of the art methods always been a driving factor of developments in machine learning, it should not be the ultimate goal of our research.*

## 3. Cherry Picking And Cherry Planting

Randomization plays an important role in the overall performance of RL algorithms both in theory and practice. For example, environments might be inherently stochastic, or in function approximation methods, e.g., deep neural networks, there is randomness in parameter initialization. In the simulated environment, the choice of *random* seed mainly determines these randomnesses, resulting in different agent's behavior.

Recently there have been discussions on why random-seed optimization might be a wrong thing to do (Henderson et al., 2018). There is a line of research that the authors run their algorithms many times and report the result of runs that their algorithms perform well. While this way of evaluating the performance of an algorithm is useful when we care about planning, this is not suitable for slightly more general goals. In general, one can refer to this way of reporting the results as *cherry picking*.

*Cherry planting*: Besides the cherry picking, there are many scientific works which go beyond the cherry picking and do cherry planting. They not only optimize their algorithm but also they adapt the environment to their algorithms. For instance, there are many simple simulation-based control environments that the experimenter can adjust the environment parameters desirable. If we also tune the environment of interested, then one could easily outperform baselines which are optimized for another set of environments parameters.

This approach and this way of doing empirical investigation is a natural outcome when the goal is to outperform the existing methods instead of providing a scientifically meaningful practical development. It is worth stating that both cherry picking and cherry planting are scientifically valid as long as the goal of the research is providing further insight in the understanding of RL problems, rather than chasing the leader-board of DRL scores.

## 4. Benchmarks

*If we aim to outperform baselines on a set of toy control problems, simple adaptive control does it all.*

The existing user-friendly simulated environments and wrappers, (Todorov et al., 2012; Bellemare et al., 2013; Machado et al., 2018; Brockman et al., 2016) for RL study made it possible to have numerous developments in RL study. Without these emulators, RL could not get the attention it deserves or even the current excitement from the practitioners, and we probably could not make the current significant progress in both empirical and theoretical RL.

Among the existing environments, the Atari and the Mu-JoCo are the most popularized ones. The most used set of MuJoCo environments in the RL community is a set of toy environment that are excellent test beds to study the behavior of RL algorithms. However, they are not practically sophisticated enough and does not deserve a significant body

of research dedicated to performing well on them. It makes much more sense to use them as useful test beds and study the behavior of RL methods. It makes much more sense to make these environments harder to see where our algorithm breaks, or make them easy to see where our algorithms are the most suitable. While competitions are necessary components of research development, our final goal should not be forgotten. It is worth noting that this chasing makes sense if we design a real-world task-specific RL agent. For example, in computer vision, researchers develop algorithms and compete to perform well on a given task, while the task is similar to the desired real-world one that we aim to solve. But competitions makes less sense if the tasks were toy vision tasks and far from practice. .

Unfortunately, a great chunk of our researchers' time has been dedicated to designing algorithm for toy MuJoCo environments with the goal to outperform the existing methods rather than principled studies, the studies which bring a useful amount of insight in developing RL methods for high dimensional real-world problems.

A similar phenomenon is also the case for Atari environment. There are quite many works and papers with extensively expensive empirical studies which clearly overfit to those environments and fail to outperform a random-uniform policy in simple two-state two-action Markov decision processes. While the RL community suffers from a lack of resources, e.g., experts' time and computation power, limiting such research types could help to allocate our focus more efficiently.

*What does it even mean to outperform an algorithm?* A wise person once said, "saying my number is bigger than yours is the lowest kind of science". In empirical RL study, mainly, we evaluate our methods on a given set of environments. We call an algorithm more significant if it has a better overall performance on these environments. However, this notion of "overall better performance" is a vague notion and the notations used in the literature are neither metrics nor transitive. Therefore not a useful tool to compare methods, putting aside why we even want to compare algorithms in the first place.

If we are given a single practical task which worth spending our time to design an efficient algorithm for, it might be fine to overfit the task. However, when we deal with toy setting where we urgently need to understand the basics of RL, then competing there has less meaning, and barely informative specially when we compare a bunch of numbers.

## 5. What We Are Even Optimizing?

In many machine learning tasks, e.g., classification and regression problems, we are given an objective function, e.g., empirical risk or a surrogate one, that we optimize over to

come up with the desired model. In contrast, in RL we do not have a clean objective function to optimize. In the online setting, we mostly need to collect our training data, requiring exploration and exploitation. In the batch setting, where we access to a set of offline data, mainly there exists no such concept as test data. In this setting, generally, we may not be able to test our agent against the real world.

If we have a good statistical understanding of our objective function, e.g., classification, climbing the leader board of state of the art corresponds to the designs of statistically-sound optimization methods which perform well in practice. In RL, when we do not have a clear understanding of our objective function yet, we better do not spend the majority of our researchers' time in the design of optimizer, instead on the study for statistically efficient RL algorithms.

## 6. Empirical And Theoretical Study

As we argued before, RL problems, by their nature are hard. Besieged this information theoretic hardness, we also argued that the research in both theory and practice are heavy to lift. Consider the process of an empirical study. For such studies, mainly, there is an initial hypothesis suggested by experienced researchers and a group of experienced programmers empirically test the hypothesis. This empirical study can also be accomplished by a group of undergraduate students who are supervised by senior researchers. Now consider theoretical studies, for this sort of researches, due to the hardness and complication of RL problems, mainly junior researchers have a hard time to accomplish a significant contribution. Therefore, it mostly requires experienced and senior researchers to spend their time on these problems to make considerable contributions.

Now consider the theorists vs practitioners ratio. The number theorists in RL compared to the practitioners is almost incomparable, while also there is a considerably week bonding between these two community. These issues further stimulate the slowness of this field. Clearly, there is a higher and urgent need for more theorists, as well as practitioners with much stronger bonding between them to boost up the advances in RL.

## 7. The Goal Of Our Research

RL is a broad topic of research and has many aspects to study. One can study RL from finding a good policy, e.g., optimal, safe. One can consider sample complexity, how many samples an algorithm takes to reach a good behavior, or how much of reward an algorithm losses due to exploration. Undoubtedly, this list can go on and on. In order to have more clear scientific contributions, our papers should make their statements clear about the problem they tackle and what are the achievements. They also need to provide

sufficient evidence to back their claims, and particularly, what are the cases they fail and they perform very well. Furthermore, in their empirical comparison, they are required to compare against the algorithm in those realms.

## 8. What is the role of reviewers?

In the last few years, we had an era that more computations and empirical studies were tickets for acceptance of papers, even best paper awards. That exciting era played a significant role in encouraging more and more empirical developments in RL, resulted in a fantastic set of empirical studies. This movement toward more empirical research stimulates the community toward appealing more empirical development. But, all of it happened at the time that we have not had reasonably good theoretical development in RL. Consequently, there is an urgent need for such advancements to serve the high interests in the empirical side.

Moreover, there exists another serious problem worth considering. As we argued before, constructing a theoretical work in RL is significantly time consuming, hard and requires an extensive set of expertise. To support this hand wavy statement, we randomly chose a famous RL theorist, and also a famous practitioner who are currently active researchers. The theorist was able to write approximately five papers in 2018, but the practitioner wrote about 50. We slightly perturbed these numbers to approximately conceal these parties' identities. We do not report these numbers as factual evidence, instead, as a piece of non-negative information to update the readers' belief or possibly narrowing down the confidence set.

Now instead, consider a less famous theorist who writes a paper. Since the principled understanding of RL is still preliminary, this work might provide a slightly better-principled understanding of RL problems. Asking this class of works to provide extensive empirical studies can be considered as a ***most damaging*** act on the development of RL [2]. Moreover, these primary and theoretical works might be hard to implement. For example in high dimensional RL, they might be statistically efficient but not computationally (Jiang et al., 2017). Furthermore, also consider that this class of works are mainly not tuned for a specific instance of simulation-based environments, rather general.

A famous example of such algorithms is the UCRL (Jaksch et al., 2010) which comes with order optimal regret upper bound guarantees on tabular Markov decision processes. But it is obvious that a simple heuristic algorithm can easily outperform UCRL on some predefined domains.

Imagine we have an Markov decision process with 10 states

$\{1, 2, \ldots, 10\}$ and 10 actions $\{1, 2, \ldots, 10\}$. Imagine the optimal decision at any state is its index, i.e., optimal action at state 4 is the 4th actions. In this case, nothing can do better than an algorithm which naively maps states to actions equal to their indices. In this case, UCRL would result in a huge regret for this instance. However, since UCRL guarantees are instance free, its regret guarantee still holds, and it did what it promised us it would do. Clearly, if we change the environment a bit, the heuristic algorithm breaks while UCRL would still behave as it promises.

The main point of this section is to convince the readers that 1) we are far from any claim about a reasonably good understanding of RL. Consider that we just recently learned how properly use Q-learning (Jin et al., 2018). Also, 2) do not expect theoretical works beyond what they promise. For instance, if a method guarantees a low variance estimation of the gradient in policy gradient and also guarantees monotonic improvement (Kakade & Langford, 2002), we should not expect this algorithm to outperform anything when the criterion is the final performance or reward. The theoretical analysis of this work just did not claim optimality. 3) Theoretical works build our understating brick by brick, and the results are constructed on the top of one another.

***We just simply may not go to the moon before understanding how to build a simple engine.***

To aim this goal, let's ***pave the road*** for the principled RL studies, both in theory and practice.

> **The final quote by a wise person**:
>
> This is a big lesson. As a field, we still have not thoroughly learned it, as we are continuing to make the same kind of mistakes. To see this, and to effectively resist it, we have to understand the appeal of these mistakes. We have to learn the bitter lesson that building in how we think we think does not work in the long run. The bitter lesson is based on the historical observations that 1) AI researchers have often tried to build knowledge into their agents, 2) this always helps in the short term and is personally satisfying to the researcher, but 3) in the long run it plateaus and even inhibits further progress, and 4) breakthrough progress eventually arrives by an opposing approach based on scaling computation by search and learning. The eventual success is tinged with bitterness, and often incompletely digested because it is a success over a favored, human-centric approach.[a]
>
> ---
> [a] www.incompleteideas.net/IncIdeas/BitterLesson.html

---

[2] As repeatedly stated, this paper, we fully agree that the empirical studies are the main axes of developments in RL

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
