# OpenReview forum: "Maybe a few considerations in Reinforcement Learning Research?"
_ICML.cc/2019/Workshop/RL4RealLife — Submitted to RL4RealLife 2019_

### Official Review · AnonReviewer1 · 2019-05-16
**A good paper with few concerns**

**Rating:** 4
**Confidence:** 5

**Review:**

The author expresses several concerns in empirical RL research, including policy gradient methods and prevailing benchmarks. The author also points out the need for more theorists. In general, it is an insightful position paper. I only have a few concerns.

1) The author defines "deep RL" as "RL using deep learning". This definition is somehow misleading. Here I quote a wise person, as the author usually does,

"The novelty in deep reinforcement learning isn't the combination of deep networks with RL methods; one of the oft-cited early successes of RL, TD-Gammon, was all about the MLP. Nor is it really in its chief algorithms: experience replay is 25 years old. Instead, it seems to me that the distinctive characteristic of deep reinforcement learning is its emphasis on perceptually complex, messy environments such as the Arcade Learning Environment, richly textured 3D mazes, and vision-based robotics. To paraphrase Rich Sutton: deep reinforcement learning is not a solution, it's a collection of problems."

2) Under this definition of deep RL, ALE and Mujoco are indeed reasonable benchmarks for deep RL algorithms, although far from perfect. Previous benchmarks (e.g., various grid worlds, mountain car) are usually not enough to test the ability to scale up (in terms of perceptual complexity) for deep RL algorithms. I agree ALE and Mujoco are just toy problems, but they are definitely in the correct direction. Climbing the leader board of them is definitely meaningless. But reaching a reasonable performance level and outperforming proper baselines on them are indeed necessary for claiming a contribution in deep RL.

"a great chunk of our researchers’ time has been dedicated to designing algorithm for toy MuJoCo environments with the goal to outperform the existing methods rather than principled studies"
I believe it is unfair to say this. There is an obvious evidence. Researchers using ALE as benchmarks usually do not compare with Rainbow. And the community does not expect they to outperform Rainbow. Similarly, deep RL researchers using Mujoco usually do not compare with evolutionary methods. They are not chasing high scores, but trying to test the scalability of their algorithms, which is indeed principled and necessary for claiming a contribution in deep RL.

"fail to outperform a random-uniform policy in simple two-state two-action Markov decision processes."
In terms of killing a fly, do you expect a gun to be better than a fly swatter?

3) Cherry planting itself is good, provided all compared algorithms are tuned with equal effort. The problem comes from researchers, not the paradigm of cherry planing itself. Like theorists provide theorems with assumptions, practitioners claim performance improvements only for specific environments. I think the author needs to rephrase the discussion about cheery planting to make it more unambiguous.

4) We definitely need more theorists. But theorists should put more effort on deep RL theory. Clearly, the theory of deep RL, deep learning and neural networks are far from mature. We need more theorists towards understanding them. The physics before Newton is now referred to as "miscellaneous earlier efforts". We do need a Newton for deep RL. Unfortunately, many are working towards more miscellaneous earlier efforts instead of Newton, which, I believe, also "stimulate the slowness of this field".

---

### Official Review · AnonReviewer2 · 2019-05-25
**Misinformed**

**Rating:** 1
**Confidence:** 4

**Review:**

To go through the paper claim-by-claim:

"[RL] is shockingly barely understood and deserves much more attention from theorists and practitioners" -- This sensationalist language overstates the problem. There is certainly more to study and understand about RL, but we also do understand quite a bit about it already. Please note that this field had been mostly theoretical until ~6 years ago, and had built on a lot of control theory, which is literally centuries old.

"They suggest that there are many settings that there might not even exist any RL method which achieves our desired performance" -- Is this a problem with our desires, problem formalization, or something else? This statement by itself is just too vague to judge whether it's valid or not.

Policy gradient methods will definitely benefit from more theoretical and practical studies, but this is well understood -- this is exactly why they are a hot research area.

The "cherry picking" section simply restates part of the issues described in the "DRL that matters" paper that it cites. The issues described in the "cherry planting" section are a problem only when the parameters chosen in the experiments aren't stated in the paper, which is again one of the issues touched upon in the "DRL that matters" paper. A more general version of this issue is that authors often try to invent their own benchmarks on which their method is the best.

Re:the benchmarks, the statement "We call an algorithm more significant if it has a better overall performance on these environments" is an oversensationalization of the issue. Personally, I have never encountered the phrase "this algorithm is more significant" in any paper. Papers typically say that the proposed algorithm beats others on several benchmarks, but that's about it.

"In contrast, in RL we do not have a clean objective function to optimize." -- This is just patently not true. Most RL methods are geared towards a very well studied family of objective functions, expected linear additive utility. "Finite-horizon reward", "discounted reward", and several others fall into this category. Are they appropriate for every setting? Certainly not, and we do need more research into efficient methods for other objectives. However, the methods that are published _try to_ optimize for extensively researched objectives (although, due to approximations, they don't usually give guarantees).

"Clearly, there is a higher and urgent need for more theorists" -- While I do appreciate theory very much, this statement is purely an opinion hardly based in fact. There is just nothing in this paper to back it up.

"To support this hand wavy statement, we randomly chose a famous RL theorist, and also a famous practitioner who are currently active researchers. The theorist was able to write approximately five papers in 2018, but the practitioner wrote about 50". -- This statistic, to put it scientifically, has a very high variance, and to put it in plain language, is purely anecdotal (and likely biased). You can surely find empirical researchers who publish 50 papers a year, but these are mostly professors with large, well-funded groups. Neither the average, nor the median, nor the mode are anywhere near this figure. At least, without describing this study's methodology, you can't claim its results as credible.

"Asking this class of works to provide extensive empirical studies can be considered as a most damaging act on the development of RL" -- There are very respectable venues where this isn't asked, and where quite a bit of RL theory is published, e.g.,  COLT. Even at ICML and NeurIPS many reviewers are sympathetic to this statement.

"we just recently learned how properly use Q-learning (Jin et al., 2018)" -- Have we, really? What does it mean to "properly" use it? Again, this kind of sensationalist language is only damaging when discussing serious matters.

In summary, I think the paper is misinformed, unfortunately. It does make some reasonable points, but mostly it is just too sensationalist, and inaccurate at that, to be taken seriously. I would suggest at the very least rewriting it to remove the sensationalist language and to make it look less like a theorist's complaint about the field of RL.

---

### Decision · Program_Chairs · 2019-05-28

Reject